# UK Women’s Views of the Concepts of Personalised Breast Cancer Risk Assessment and Risk-Stratified Breast Screening: A Qualitative Interview Study

**DOI:** 10.3390/cancers13225813

**Published:** 2021-11-19

**Authors:** Charlotte Kelley-Jones, Suzanne Scott, Jo Waller

**Affiliations:** Faculty of Life Sciences & Medicine, King’s College London, London SE1 9RT, UK; suzanne.scott@kcl.ac.uk (S.S.); jo.waller@kcl.ac.uk (J.W.)

**Keywords:** breast screening, mammography, breast cancer, risk stratification, personal risk, risk assessment, acceptability, screening preferences, overdiagnosis, screening harms

## Abstract

**Simple Summary:**

Risk-based breast screening will involve tailoring the amount of screening to women’s level of risk. Therefore, women at high-risk may be offered more frequent screening and over a longer period of time than those at low risk for whom less screening may be recommended. As this will involve considerable changes to the NHS Breast Screening Programme, it is important to explore what women in the UK think and feel about this approach. Analysis of in-depth interviews revealed that some women would find both high and low-risk screening options acceptable whereas others were resistant to the prospect of reduced screening if they were assessed as low-risk. We also found that the idea of risk-based screening had little influence on the attitudes of women who were already sceptical about breast screening. These findings highlight the communication challenges that will be faced by those introducing risk-based screening and suggest a need for tailored support and advice.

**Abstract:**

Any introduction of risk-stratification within the NHS Breast Screening Programme needs to be considered acceptable by women. We conducted interviews to explore women’s attitudes to personalised risk assessment and risk-stratified breast screening. Twenty-five UK women were purposively sampled by screening experience and socioeconomic background. Interview transcripts were qualitatively analysed using Framework Analysis. Women expressed positive intentions for personal risk assessment and willingness to receive risk feedback to provide reassurance and certainty. Women responded to risk-stratified screening scenarios in three ways: ‘Overall acceptors’ considered both high- and low-risk options acceptable as a reasonable allocation of resources to clinical need, yet acceptability was subject to specified conditions including accuracy of risk estimates and availability of support throughout the screening pathway. Others who thought ‘more is better’ only supported high-risk scenarios where increased screening was proposed. ‘Screening sceptics’ found low-risk scenarios more aligned to their screening values than high-risk screening options. Consideration of screening recommendations for other risk groups had more influence on women’s responses than screening-related harms. These findings demonstrate high, but not universal, acceptability. Support and guidance, tailored to screening values and preferences, may be required by women at all levels of risk.

## 1. Introduction

The UK National Health Service Breast Screening Programme (NHSBSP) currently invites all women aged 50–70 in the UK for triennial mammography. Since its initiation in the late 1980s, scientists have gained knowledge about the heterogenous nature of breast cancer [1] and identified ‘classic’ risk-factors additional to age, including family history, reproductive and hormonal history and lifestyle which can be used to predict individual levels of breast cancer risk. More recently, risk-prediction models have integrated mammographic breast density and polygenic risk scores which have increased their discriminatory efficacy [2,3]. This has led to an increased interest in risk-stratified breast screening (RSBS), i.e., adapting screening recommendations to a woman’s breast cancer risk [4]. Simulation models suggest this approach has the potential to increase the benefits of screening (via early detection of cancer) while decreasing screening-related harms, e.g., false positive results and overdiagnosis [5,6,7,8]. However, the incorporation of RSBS within an existent screening programme will require major shifts in the way that screening is delivered and necessitate the involvement and agreement of all relevant stakeholders [9,10]. The screening preferences and values of prospective service users and their willingness to engage with this screening approach are of key importance.

High uptake of multifactorial personal risk assessment (PRA) is fundamental to the feasibility of RSBS. Survey studies have demonstrated that women broadly support the prospect of undergoing PRA and would be willing to receive risk feedback [11,12,13,14] as an opportunity for greater perceived control over their level of risk and to reduce levels of breast cancer worry [12,14]. Nevertheless, the Predicting Risk of Breast Cancer at Screening study (PROCAS) reported relatively low uptake of PRA, at 38% [15]. Therefore, there is a need to explore facilitators and barriers to PRA in more depth.

Focus-group studies have found that women’s concerns about PRA include cognitive burden, low self-efficacy [16,17] and emotional threat [16,17,18,19]. However, researchers have generally paid less attention to PRA than RSBS and have only recently started to quantitively investigate the relative acceptability of individual risk measures (e.g., lifestyle versus polygenic risk) [12,13].

Previous research has consistently found high levels of support for shorter screening intervals for women categorized as high-risk in a risk-stratified screening programme [13,14,17,18,19,20]. As enthusiasm for breast screening is well-documented [21], it is possible that women would welcome increased screening almost regardless of PRA feedback [13,20]. By contrast, evidence suggests that women are resistant to any reduction of screening for those at lower-risk, with some interpreting this as a form of healthcare rationing and demonstrating limited appreciation of the need to minimize screening-related harms [16,17,18,19]. However, it is possible that this is not the full picture as research has mainly focused on women’s attitudes to risk-based screening frequency rather than exploring this feature in conjunction with variations in other aspects of screening, such as age-range of eligibility and number of risk groups (for instance five risk groups: ‘very low’, ‘low’, ‘moderate’, ‘high’, ‘very high’, as opposed to three: ‘low’, ‘average’, ‘high’). Furthermore, few studies [11] have investigated the responses of women who are either too young for, or have chosen not to attend, breast screening. As these women may have different attitudes to PRS and RSBS compared to those with breast screening experience, it is important to include their perspectives.

The present study explored these gaps in the literature using in-depth interviews with UK women currently at, or approaching, the age of NHSBSP eligibility to:Explore women’s understanding and experience of breast screening in the context of their responses to PRA and RSBS;Gain insight to women’s understanding of and prospective willingness to undergo multifactorial PRA;Explore prospective acceptability of possible RSBS scenarios in which screening frequency, age-range of eligibility and number of risk groups might vary.Assess how risk-management options, such as lifestyle change and chemoprevention, are seen in the context of RSBS scenarios.

## 2. Materials and Methods

### 2.1. Participants

A purposive sampling frame was devised with the aim of recruiting a sample from a range of age groups (40–49; 50–59; 60–70); socioeconomic backgrounds (occupational social grades ABC1 and C2DE: http://www.nrs.co.uk/nrs-print/lifestyle-and-classification-data/social-grade/, accessed on 1 April 2020) and breast screening experience (pre-eligible, regular attender, occasional attender, non-attender). As our focus was on women at population-level breast cancer risk, those with a previous and/or current diagnosis of breast cancer were excluded alongside women in receipt of RSBS under NICE guidelines for high familial risk (https://www.nice.org.uk/guidance/cg164, accessed on 1 April 2020). Participants were recruited by a recruitment agency, Saros Research Ltd. (https://www.sarosresearch.com, accessed on 30 April 2020), which has a large participant panel from which women were recruited in three stages: (a) Identification of participants through an initial on-line screener tailored to the sampling criteria; (b) Telephone screen to check study eligibility and participants’ understanding of what would be involved; (c) Dissemination of relevant recruitment materials and request to sign an informed consent form. Although we did not specify a sample size a priori, 20 women were initially recruited followed by a further five whereupon it was felt that little new information was forthcoming. Participants gave written and verbal consent prior to the interviews and received a payment of GBP 50 for their time, in line with Saros’ standard procedures. The study was approved by the King’s College London research ethics committee (Ref: LRS-19/20-17949).

### 2.2. Procedure

As we were interested in individuals’ decision-making and there was a considerable amount of information about RSBS to convey, an interview study design was used, rather than focus groups, to ensure that information provision could be tailored to participants’ needs. Prior to interviews, women were sent a participant information sheet and asked to complete a short questionnaire which included sociodemographic and psychosocial measures associated with breast screening behaviour: age, educational level, ethnic background, perceived breast cancer risk, and two items regarding the intensity and frequency of breast cancer worry from the validated Cancer Worry scale [22]. However, this information was used to describe the sample rather than for the purposes of grouping participants for analysis. Online semi-structured interviews, (mean duration = 73 min; range: 55 to 115 min), using in-depth interviewing techniques [23] were conducted via Microsoft Teams between September and October 2020. All interviews were carried out by the first author (C.K.J.), a female PhD student in the same age-range as the participants, who has completed training in in-depth interviewing techniques. Interviews followed a topic guide arranged in three parts (see Appendix A for a summary of the interview topic guide):

Part 1: In line with our first aim, this section explored women’s attitudes towards breast cancer; their understanding and awareness of the current NHSBSP, and their perceived risk and knowledge of breast cancer risk factors.

Part 2: As outlined in the second study aim, this section was designed to gain insight to women’s cognitive and emotional responses to PRA. Participants were asked about their willingness to undertake a blood/saliva test and mammogram to determine PRS and breast density, respectively. Additionally, they were asked about their willingness to self-report details of family history of breast cancer, health-related behaviours and reproductive and hormonal history as part of PRA. They were then asked to comment on the idea of combining the risk information into a personal level of risk and willingness to receive risk feedback.

Part 3: In accordance with the third study aim, this section introduced women to RSBS and explored their responses to screening scenarios defined by risk-based variations in age range of screening eligibility, screening frequency and number of risk groups. We opened with a short presentation outlining the benefits and harms of age-based breast screening and how personal risk information could be used to improve the NHSBSP by allocating women to risk groups with tailored screening recommendations. To avoid cognitive overload and enhance understanding, participants were then introduced to RSBS by way of two visualisations (see Figure 1), designed for the study, with embedded animations used to present risk-modified screening attributes so as to build a picture of how this possible screening approach may look in practice in four steps:

(i) Risk-based variations to start and stop ages of breast screening with participants asked to anticipate how they would respond if categorised as high, average or low-risk, and offered screening from 45–75 years, 50–70 years and 55–65 years, respectively (see Figure 1a).

(ii) Screening intervals adjusted to level of risk alongside age-range of screening eligibility with participants invited to imagine how they would respond in each scenario, e.g., screening every 12 to 18 months from age 45 to 75 years for high-risk and every 5 years from age 55 to 65 years for low-risk (see Figure 1a).

(iii) It was explained how more precision could be gained by including an additional risk group (moderate risk see Figure 1b) and changing the status of the low-risk group to ‘very low risk’. The screening attributes of age-range and screening frequency were then re-introduced, and women were asked to respond to each RSBS scenario in turn as in step (ii).

(iv) Finally, and in line with the fourth study aim, the interviewer explained that breast screening was only one way of managing breast cancer risk and introduced two risk-reducing/preventative options (lifestyle change and risk reducing medication) to ascertain whether these had any impact on the prospective acceptability of the suggested RSBS scenarios (see Figure 1b). Finally, the harms of breast screening were reiterated to assess whether these influenced responses to low-risk screening scenarios.

### 2.3. Data Analysis 

Interview transcripts were analysed using Framework Analysis [24] which is a form of thematic analysis. Framework Analysis was developed in an applied policy research context to provide an accessible analytic process designed to achieve answers to specific objectives. It is defined by a matrix-based analysis which allows researchers to systematically and comprehensively synthesise a large data set for analysis by case and code. Importantly, it offers the ability to compare data across cases (interviewees) and within-cases (interviews) to identify any commonalities or differences in participant responses. Although Framework Analysis is not bound within any epistemological approach, we adhered to a contextual interpretation of the data. Therefore, the interviewer’s reflexive engagement, as a regular attender of NHS breast screening herself, is acknowledged as a resource for knowledge [25].

Three interconnected analytical stages central to Framework Analysis were followed. Initial familiarisation with the transcripts was followed by an iterative process of identifying codes (C.K.J.) which were refined through discussion until we had developed and agreed a coherent coding index and preliminary themes (C.K.J., J.W., S.S.) enabling the assignment of coded data to matrices using the ‘framework’ feature in NVivo 12™ (QSR International). The coded data were then summarised allowing us to synthesise the data set and refine the thematic framework (outlined in Appendix A). Finally, the data were interpreted through the close inspection of each theme which allowed us to draw on commonalities and/or differences between anticipated cognitive and emotional responses to PRA and RSBS scenarios. We followed the COREQ reporting guidelines (see Appendix A for COREQ checklist).

## 3. Results

### 3.1. Participant Characteristics

Interviews were conducted with 25 women with a mean age of 55 years (range 40–68). Women had a range of breast screening experiences: pre-eligible (*n* = 7); regular attenders (*n* = 11); occasional attenders (*n* = 3) and non-attenders (*n* = 4) and were evenly split between higher and lower occupational social grade groups. Nearly half the sample were educated to university level. Three-quarters were from white British backgrounds. More than a half perceived their breast cancer risk to be the same, or lower, compared to others in their age group, and only moderate to low levels of breast cancer worry were reported (see Table 1).

During the interviews, two women described a close family member with breast cancer and others had personal experience of friends and colleagues with the disease. All participants were aware of the NHSBSP and were able to identify its main benefit in terms of early detection. However, many women, including regular screening attendees, had little awareness of current NHS screening intervals and/or age-range of screening eligibility. In terms of harms, women cited radiation and pain of mammography, with fewer mentioning false positive results and risk of overdiagnosis.

### 3.2. Thematic Framework

Five themes were identified in the analyses which are summarised in Table 2. Each of these themes are described below with illustrative quotes. Details in parentheses following the quotes represent participant ID; screening experience (pre-eligible; regular; occasional; never attended) and age in years.

#### 3.2.1. Risk Perceptions and Acceptability of Personalised Risk Assessment

##### Perceived Risk

Most women believed their breast cancer risk to be average, or lower, as they had no family history and adhered to healthy lifestyle behaviours. Some of those over 60 considered themselves to be a ‘safe bracket’ due to their hormonal age. Where risk was perceived to be higher than average, this was due to first-degree family history, poor health status and in vitro fertilisation treatment. 

##### Acceptability of Providing Risk Information

Overall, women would be willing to provide information about family and reproductive history, and many now expect to be asked questions about health behaviours:

“I think that’s so routine, isn’t it? You’re asked it on every questionnaire, “Do you smoke, drink?” (4: Regular, 50 years).

Responses to the idea of tests to assess polygenic risk were more varied with some suggesting that this would be: “… a no brainer for me. I’d be like, “Yes, where do I sign” (1: Pre-eligible, 48 years).

Others were more reticent:

“I think that would worry me, if it came back that there were enough little pointers to the direction that I could develop breast cancer …. Oh, I don’t know the answer to that one” (16: Regular, 67 years).

Although most women indicated they would agree to having their breast density assessed, some non-attenders were concerned by the need for mammography due to concerns about radiation. One woman who had made the decision not to accept an invitation for screening due to her awareness of screening-related harms and her antipathy towards social pressure to attend breast screening would question an assessment of her breast density:

“Well, it would still be a mammogram, wouldn’t it? For all intents and purposes, it’s getting involved in the system” (9: Non-attender, 51 years).

##### Understanding of PRA

Women seemed to understand the idea of combining individual risk measures to compute an overall level of personal risk and appreciated the rationale for this in terms of their perceived limitations of a ‘one size fits all’ screening model:

“So just because you’re female and got boobs doesn’t automatically mean you’re at risk” (16: Regular, 67 years).

Some interviewees acknowledged the trending application of risk-profiling and were able to anticipate RSBS:

“These days everything is so tailored and personalised and we’re in that kind of society now…we can’t have that standard one size fits all model anymore” (11: Pre-eligible, 46 years).

“And then they can go on from that to decide whether you need more or less breast screening and so it’s in everybody’s interests” (21: Regular, 51 years).

Several participants raised concerns about the modifiable nature of certain risk factors and, therefore, identified the need for PRA to be organised on a rolling rather than one-off basis:

“So, if I had my risk assessment done today. And then next week I’m suffering terribly with hot flushes…and suddenly take HRT, would that be flagged to the risk-assessment people?” (15: Regular, 59 years).

##### Willingness to Receive Risk Feedback

Most women indicated that they would wish to receive PRA feedback to provide certainty and an opportunity to be pro-active:

“There’s nothing worse than sort of like having that wondering. At least if you know… you can deal with it, however hard that might be” (21: Regular, 51 years).

Others recognised the potential of PRA to help health professionals to monitor their breast health more effectively over time:

“Because then if you find something, and you think ‘I’m in that 10% of really low, look what I’m finding’… so they’re [health professionals] are going to think what’s changed. What’s happened, what’s she doing differently? So, they can look at the information they hold on me to see why this change has occurred?” (20: Regular, 52 years).

Women’s willingness to receive risk feedback was also interpreted as a means of taking personal responsibility for risk management:

“Well, there’s a risk in knowing itself … because of the anxiety from it. But you can’t just plod along thinking everything’s fine… it’s just being responsible for your health in a way …. It’s just something that you’d have to do whether it’s pleasant or unpleasant” (18: Pre-eligible, 40 years)

However, a small number of women misinterpreted PRA as a diagnostic procedure which was reflected in their reasons for receiving risk feedback. One woman suggested that it would allow her to plan a bucket-list, and another said she would go to Germany to have an operation. Such misconceptions may have played into some of their concerns about the potential negative impact of risk feedback on women’s emotional health and quality of life:

“… if you’re given a risk profile and it’s high and you then realise you’re at a stage when you’re quite terminal, I don’t know whether I’d really actually want to know or whether I’d just want to live my life without knowing if nothing can be done” (4: Regular, 50 years).

Some women questioned whether they would be able to choose to engage with PRA and, if so, “what happens if people refuse the risk assessment? Where do they fall into the categories?” (15: Regular, 59 years). Other participants presumed PRA would be mandatory and that women would adjust to this over time:

“I mean this is going to be a mandatory kind of thing. You’ll have to go to the doctor’s to give this information, and it’ll just be done, as a matter of fact, in the future” (6: Regular, 56 years).

#### 3.2.2. Ways of Responding to RSBS Scenarios

##### A Typology: (1) ‘Overall Acceptors’; (2) ‘More Is Better’; (3) ‘Screening Sceptics’

Women considered what RSBS could look like with the adaptation of screening frequency and age-range of eligibility to personal level of risk (see Figure 1a,b). We identified three types of response. The ‘overall acceptors’ (*n* = 10) considered most aspects of RSBS to be acceptable albeit with some conditions. By contrast, women who thought ‘more is better’ (*n* = 9) supported high-risk scenarios where increased screening was proposed but largely rejected reduced screening for those at low-risk. ‘Screening sceptics’ (*n* = 6) were characterised by ambivalent and negative attitudes towards breast screening, and tended to find aspects of increased screening at higher risk unacceptable. Table 1 details participant characteristics by each response type. These three types of responses to the ways in which screening might be adapted to risk level are detailed below.

##### Age-Range of Screening Eligibility

The ‘overall acceptors’ thought that most women profiled as high-risk would welcome an extended age-range of screening eligibility as a source of reassurance and increased perceived control. In most cases, any concerns regarding the discomfort of breast screening were countered by the perceived benefits of increased surveillance:

“I think … you would want to have those additional checks and those earlier checks as well, because obviously that’s going to be a concern if you’re high-risk and you’re going to want to do something about it, so I think yes, definitely” (18: Pre-eligible, 40 years).

Likewise, the ‘more is better’ group also supported this prospect but on the basis of ‘how it should be’ in an ideal world:

“I mean, it’s always down to cost anyway. So, if they can’t increase it because of the cost, then I do think that the age-range is acceptable” (22: Regular, 60 years).

Generally, ’screening sceptics’ understood the rationale for this as, for example, “more individualistic …rather than just lumping them. So, it gives a wide parameter of risk between 40 and 75” (24: Non-attender, 65 years). Nevertheless, many would not regard this as personally acceptable as they were either in their late 60′s and/or had already made the decision not to attend breast screening. 

The ‘overall acceptors’ and ‘screening sceptics’ largely accepted shortening the age-range of eligibility for low-risk groups as an appropriate re-allocation of NHS resources and to avoid any unnecessary screening. They were also reassured that they would still be invited for screening during the period of menopausal hormone changes, but some did suggest that they would need to know that this recommendation was evidence-based:

“As long as it’s backed up by heaps of scientific data and everyone agrees that 55 to 65 is when everybody should really be checked, because that’s a scare time, but as long as it all stacks up then it makes perfect sense to me” (5: Regular, 59 years).

By contrast, women who thought ‘more is better’ disputed this scenario on the basis of knowing women younger than 55 to be diagnosed with breast cancer and perceiving low-risk as “still a risk with the menopause, there’s everything that’s going on” (1: Pre-eligible, 48 years). One woman in this response category indicated that she would find a shorter age-range of screening eligibility difficult due to a perceived pervasiveness of breast cancer:

“I think that even though I am low risk, my social circle, or the women that I’m in touch with or the media, all of those things, because breast cancer is such a common thing now, I would worry that… because my risk profile has given me a low-risk, [screening from] 55 seems too late” (4: Regular, 50 years).

##### Screening Frequency

In terms of reducing screening intervals for women at high-risk, ‘overall acceptors’ indicated that they would adopt a ‘grin and bear it’ attitude underscored by an understanding of clinical need and the perceived benefit of peace of mind. Even 6–12-month intervals would be valued as a source of reassurance as “…a lot could happen in 12 months. I mean a lot could happen in six months. But you know, to some people [12 months] might as well be five years if they’re at high-risk” (20: Regular, 52 years). However, some ‘overall acceptors’ expressed concerns that 6–12 months would be intrusive and regarded 12–18 months as more manageable. In some cases, resistance towards 6–12 months was tempered by the introduction of risk-reducing and/or preventative options. For instance, one woman who thought that she would opt for a double mastectomy rather than undergo this intense screening regime changed her view when informed of risk-reducing options which may allow her to reduce screening frequency:

“I mean, it’s still bad … but you can do something about it... it’s not ever going to go… but if there’s a lifeline thrown at you like a lifestyle change and maybe tablets then yes, I would take that “ (15: Regular, 59 years).

Women who thought ‘more is better’ considered both 12–18 and 6–12-month intervals acceptable and a vital source of reassurance as “anyone at high-risk should be screened as much as possible” (1: Pre-eligible, 48 years). Even where 6 months was perceived as a little excessive it would be worth it:

“It certainly seems a lot, but … if you fall into that category, I think that you’d want to know that everything was being done. So, six months does seem a little excessive to me. But, you know, if it just saves one woman, then to me it’s worth the risk” (23: Regular, 60 years).

However, this participant could understand how others would find this screening regime intrusive.

Concerns regarding the intrusiveness of high-risk screening recommendations were voiced loudly by ‘screening sceptics’ in terms of negative emotional impact and described by one interviewee as a “shadow”. Others echoed this response with concerns about more frequent breast screening making breast cancer the defining feature of women’s lives:

“I would imagine as soon as you’ve gone for one, you’re pretty much on top of another one” (23: Regular, 60 years).

“That would be a difficult adjustment to make. I’d have to be told “This is going to save your life really”, to take that on board. It would also feel like a bit of a shadow… It’s sort of not a death sentence, but you know what I mean? … it would be hard not to think negative things about that” (17: Occasional, 61 years).

The prospect of extending screening intervals to 5-years for women at low-risk was considered reasonable by ‘overall acceptors’ and some with negative perceptions and/or experience of the screening procedure thought it would be a relief:

“If I was found not to be at great risk, it would put me out of my misery having to go for it because it’s not nice” (16: Regular, 67 years).

However, these interviewees did outline concerns and conditions such as a need to trust the process of PRA and knowing how to remain vigilant during an extended screening interval:

“I suppose it’s about making sure, even in the low-risk bracket, you know what things you could be doing yourself to monitor things as well” (14: Pre-eligible, 47 years).

Most women who thought ‘more is better’ deemed the prospect of 5-year screening intervals unacceptable as this would evoke worry alongside the added burden of taking personal responsibility for checking their breasts:

“After the first year of the screen I will worry … [and will have] to wait another four years. So that means I have to check my body more frequently myself” (12: Regular, 68 years).

Several women with ’more is better’ responses thought extended screening intervals would be more acceptable if there was an option for elective screening if women are concerned:

“There should be maybe an option that even if you are in a low risk … you should be given the opportunity to have them more frequently” (2: Pre-eligible, 41 years).

Some women who thought ‘more is better’ suggested that those who found this low-risk scenario acceptable would be tempting fate, with one woman suggesting that “people are going to die” (1: Pre-eligible, 48 years). By contrast, ‘screening sceptics’ welcomed the prospect of less frequent screening and were generally more prepared to play what one woman described as ‘gamblers’ odds’ by taking personal responsibility for their health “as low-risk doesn’t mean you’re completely in the clear but just look after your health basically” (17: Occasional, 61 years).

##### No Screening—Very Low-Risk

‘Overall acceptors’ had mixed responses to the prospect of foregoing screening if at very low-risk. Although some considered it reasonable and rational, acceptability was conditional on clearly sign-posted self-referral pathways for screening. There were also indications of an endowment effect with some ‘overall acceptors’ who were regular attenders suggesting that although they would find it personally unacceptable, they could see future generations of women thinking otherwise. In some cases, where understanding of the rationale for RSBS was good, screening-related harms were reiterated, which elicited an attitudinal change from finding no screening unacceptable towards a more open-minded approach:

“So, if the health professional reassured me, I would probably be relieved …if you’ve gone through everything from the risk assessments and highlighted all the positives and negatives, and if the negatives are going to outweigh the positives, I would then be persuaded not to have the screening” (11: Pre-eligible, 46 years).

By contrast, women who thought ‘more is better’ dismissed the prospect of no screening. Moreover, the reiteration of screening harms had little impact here, with one woman stating that she didn’t mind the harms, and that if the NHS weren’t going to provide screening, she would go private and “pay for reassurance” (12: Regular, 69 years). ‘Screening sceptics’ found no screening more aligned with their current screening choices. One woman, whose decision not to attend screening had been informed by an awareness of screening-related harms, considered this option “refreshing” and hoped it could become “normalised” (9: Non-attender, 51 years).

#### 3.2.3. Influence of ‘Ladder of Risk’ on Responses to RSBS Scenarios

It was evident that screening scenarios for moderate and average risk were used by women as reference points when formulating their responses to high- and low-risk options. Moreover, women indicated that it would be useful to receive their personal risk feedback and screening recommendations as ‘a ladder of risk’ as this may reflect how they evaluate their level of risk:

“… the human psyche does weigh things up and you look at who’s above you and below you … placing yourself on a sort of scale” (17: Occasional, 51 years).

An awareness of the screening options for other risk groups would help them to ‘paint the bigger picture’ and contextualise the ‘top-heavy’ allocation of resources vis-à-vis their personal level of risk: “otherwise its meaning is cloaked really isn’t it?” (25: Non-attender, 64 years).

Women seemed to anticipate a more intense screening regime at moderate-risk and the presence of a higher-risk group above informed women’s positive responses to this screening scenario:

“Yes, now, compared to the high-risk, it [moderate risk 12–18-month intervals] seems more manageable” (11: Pre-eligible, 46 years).

Although some women would feel relief to be assessed as moderate-risk rather than high-risk, an awareness of at least two risk groups below them may increase their motivation to engage with risk-reducing options. Furthermore, the presence of two risk groups above helped women, including some women who thought ‘more is better’, to accept the proposed reduction of screening frequency and periodicity for average-risk:

“I think having seen… that there are people a lot more at risk than you, this is okay” (22: Regular, 60 years).

“seeing … there’s a couple of levels above me, I’ve nothing to worry about. It might give people a false sense of security” (16: Regular, 67 years).

Generally, women made fewer references to the wider ladder of risk when articulating their responses to low and very low-risk screening scenarios. However, in cases where this was so, women indicated that it would help them understand the rationale for less screening as a reasonable allocation of resources:

“I’d be thinking, well, there’s people that are at higher risk that need to be getting that extra level of care, and I don’t need it” (21: Occasional, 51 years).

#### 3.2.4. Concerns and Conditions of Acceptability

##### Information Support

Women across all response types raised more concerns and conditions of acceptability for RSBS than PRA. Not least, the need for RSBS to be effectively communicated in order for women to understand why changes are being implemented at all levels of risk:

“If things were explained properly, then I think the majority of women will actually go for it … there’s lack of information even now, isn’t there? (21: Occasional, 51 years).

Moreover, health professionals may be required to be “more trained and clued up” (11: Pre-eligible, 46 years) to provide information and guide informed screening decision-making “as they have a lot of work to do on this front in terms of informing and supporting women” (15: Regular, 59 years).

##### Breast Awareness Support

Women expressed lack of self-efficacy in terms of how to check their breasts and know what signs they should look out for. Consequently, the need for breast awareness training was frequently cited as a condition of acceptability for low-risk screening recommendations:

“I would be happy to come out of a risk assessment and deemed to be very low-risk, but I’m not convinced with my own approach to self-checks” (18: Pre-eligible, 40 years).

Several women thought that health professionals should be available to carry out clinical breast examinations to reassure low-risk women during the extended screening intervals:

“I would just want to see much more information about how you can examine your breasts or go to a nurse or doctor to just have them feel your breasts without going to a mammogram if you’re anxious …” (17: Occasional, 61 years).

##### Integration of PRA and RSBS

Alongside reassurance concerning the accuracy of PRA, women raised concerns around how, and by whom, any changes to their risk profile would be ‘flagged’ to screening organisers on a timely basis:

“I assume [PRA] is an ongoing thing isn’t it? Because down the line they will have, once they’ve done the screening, more results to see how accurate it can be and how it can be beneficial” (6: Regular, 56 years).

In some instances, reassurance about the efficient integration of PRA with breast screening was cited as a condition of acceptability for low-risk screening scenarios: “Okay. I’ll go with it [reduced screening frequency]. So long as there’s this risk assessment alongside it” (15: Regular, 59 years).

#### 3.2.5. Perceived Effectiveness: RSBS vs. Current NHSBSP

Despite their concerns and conditions, most participants perceived RSBS to be an improvement over the current age-based breast screening:

“It really makes sense with all that added information and you can only get better treatment… the old system seems a bit dated” (11: Pre-eligible, 46 years).

One respondent thought that this ‘old system’, i.e., age-based breast screening now appeared to be lacking:

“I don’t see where those high-risk groups, especially, because that’s where the focus needs to be, they’re just flowing through at the same rate and that could be where there’s a lot of issues” (18: Pre-eligible, 40 years).

Women also recognised the rationale to ‘modernise’ the NHSBSP in alignment with ‘exponential’ medical advance and scientific technologies, e.g., genomic profiling. On the other hand, the longevity of the NHSBSP was also regarded as a source of reassurance:

“… you know, thousands and thousands of women have been going and loving it for years and years. That’s kind of reassuring from that aspect” (11: Pre-eligible, 46 years).

## 4. Discussion

This work introduced women to PRA and RSBS and explored their thoughts and feelings about the prospect of incorporating this approach within the NHSBSP. To date, there has only been one qualitative interview study assessing the acceptability of this novel screening approach amongst UK women [26]. As McWilliams and colleagues aimed to explore women’s perceptions of extended screening intervals having been identified as low-risk through their participation with the BC-Predict study [27], this is the first study to provide a more comprehensive overview of how women think and feel about the prospect RSBS as a whole. Overall, women considered PRA and RSBS to be a good idea and appreciated the scientific rationale for this in terms of aligning the NHSBSP with scientific and technological advances, yet acceptability was subject to specified conditions.

High levels of motivation were observed for self-reporting lifestyle, reproductive, hormonal and family history information. Furthermore, most women welcomed the prospect of genetic testing which is consistent with UK and European surveys suggesting the acceptability of introducing polygenic risk tests [13,20]. Although, concerns have been raised about the psychosocial and data security implications of genetic tests, we found study participants to be more pragmatic about such concerns relative to some international focus-group studies [16,17,18,19]. With the exception of women who had chosen not to attend screening, most participants would support having their breast density assessed through an initial mammogram. However, some were keen to know at what age PRA would be implemented, whether it would be repeated and whether breast density would be assessed alongside other personal risk information. Although policy for population PRA has yet to be determined, this study demonstrates that a woman’s age and the timing of risk assessment procedures may influence the acceptability of PRA especially if mammographic breast density is to be integrated within the process [28,29].

Overall, participants would be willing to receive risk feedback as an opportunity for proactive risk-management which corresponds to findings from UK and European survey studies [11,14,30]. However, some women misinterpreted the PRA process as a diagnostic procedure with ‘high risk’ interpreted as meaning advanced cancer. This reflects the need to clearly inform women about the rationale for, and implications of PRA, in order to avoid panic in response to high-risk feedback. Women also raised concerns around whether PRA would be mandatory and questioned the screening implications for women who choose not to undertake PRA. Therefore, any future RSBS policy will have to consider the practical and ethical concerns of respecting an individual’s right not to know their risk whilst ensuring equitable access to breast screening [31].

Additional concerns and conditions of PRA included women’s expressed need to trust the reliability of their risk assessment with some indicating that they would need to personally verify the scientific evidence. Moreover, we found that women may require the support of health professionals to explain how their risk level was assessed, which risk factors contribute most to their overall risk score and what actions can be taken to reduce their risk. This is consistent with women’s preferences to receive risk feedback, especially if high, via consultation with a medical specialist [13,14,19]. Several women expected modifiable risk measures to be assessed on a regular basis to feed into screening recommendations and were keen to understand how this would happen.

As we explored RSBS as a whole, rather than focusing on a specific risk level [26] or ethnic group [32], we were able to identify three distinct responses to the concept. Previous research has demonstrated significantly greater acceptability for high-risk over low-risk screening options [11,13,14]. However, we found this may be less clear-cut than has previously been suggested, i.e., there may be limits to how much screening women are prepared to endure. For instance, some ‘overall acceptors’ would prefer 12–18 over 6–12-month screening intervals for high-risk groups as they expressed concerns about the manageability of the latter screening regime. Nevertheless, in several instances, 6–12 months became more acceptable once women were informed about the potential to manage their level of risk, and perhaps reduce the frequency of screening, through lifestyle change and/or chemoprevention. Therefore, it will be important to emphasise non-screening risk management options as a means of providing women with a sense of control over their risk level. This may also help women who think ‘more is better’ to moderate their enthusiasm for frequent mammography as a means of ‘taking care of’ their risk of developing breast cancer.

The ’overall acceptors’ were able to rationalise low-risk screening recommendations as a reasonable allocation of NHS resources. This is consistent with UK surveys demonstrating that between 53% and 58.5% of women would consider a screening interval of 4–5 years if they were assessed as low-risk [14,20]. Our findings also correspond to McWilliams et al., who found the acceptability of less frequent screening at low-risk to be contingent to perceived accuracy of PRA and evidence-based low-risk screening recommendations [26]. Although ’overall acceptors’ had mixed responses to foregoing screening if assessed as very low-risk, some thought this option would be acceptable with individual guidance concerning the balance of screening benefits and harms and/or were able to perceive this as more acceptable for younger women un-endowed with NHSBSP experience. This corresponds to previous research finding some, albeit reduced, support for no screening at the lowest level of risk [13,14]. By contrast, women who thought ‘more is better’ had negative responses to low-risk scenarios. Although their resistance did not seem to be explained by higher perceived risk or personal experience of breast cancer, there may have been an element of breast cancer worry as two women in this response group had referred themselves to be screened prior to reaching the age of screening eligibility. Furthermore, many expressed low perceived control and heightened perceptions of the severity of breast cancer as an indiscriminatory ‘killer’. Additionally, as most women who thought ‘more is better’ were regular screeners, there may have been an element of loss-aversion or sense of endowment [33].

Women’s responses to RSBS scenarios appeared to vary according to some socio-demographic characteristics, although statistical comparisons between response types were not appropriate with our small and unrepresentative sample. ’Overall acceptors’ and women who thought ‘more is better’ appeared to be similar in age, but the latter group tended to have lower levels of educational attainment (see Table 1). As survey studies have demonstrated an association between lower levels of education and negative responses to reduced screening frequency for women at low-risk [5,13], it is possible that a lack of understanding may have influenced women’s negative attitudes towards low-risk screening options. Nevertheless, this will need to be confirmed by quantitative research with a larger and representative UK sample. ‘Screening sceptics’ were educated to a similar level as ’overall acceptors’, but older than the other groups. Although these women perceived the merits of risk stratification, they remained committed to their general antipathy towards breast screening throughout the interview. Therefore, both women who thought ‘more is better’ and ‘screening sceptics’ may be indicative of a general resistance to aspects of RSBS which could be problematic at population-level [33].

By describing risk stratification scenarios with up to five risk groups, more than in previous studies [11,14], we gained insight into how women used the screening recommendations of other risk groups to formulate their responses. Moreover, women themselves indicated that it would be useful to present their personal risk feedback and screening recommendations within a spectrum of risk to help them contextualise their personal level of risk. Furthermore, the potential to move within a hierarchy of risk [34] may increase women’s motivation to engage with risk management options. Therefore, it may be useful to contextualise PRA and risk-reducing recommendations within a wider ladder of risk.

Although we outlined screening harms in more detail that previous studies [12,13,19,20,21], women still required some prompting to consider these when formulating their responses to screening scenarios, especially low-risk screening options. Although UK women’s lack of awareness and understanding of screening harms has been widely documented [35], the need to address this will be crucial to the wider acceptability of RSBS [26].

In regard to the feasibility of incorporating RSBS within the NHSBSP, this study is promising in terms of the overall acceptability of high-risk screening options and the attitudinal shifts of ’overall acceptors’ towards high and very low-risk screening scenarios. Nevertheless, our findings are consistent with survey studies reporting significant proportions of women who find low-risk options unacceptable [11,13,14,20]. As this study may be influenced by self-selection bias, we cannot dismiss the possibility that there may be many UK women whose attitudes towards screening are aligned with women who thought ‘more is better’. Therefore, more research is required to explore the resistance to de-intensified screening for low-risk women and how the clinical rationale for this can be effectively communicated against a persuasive backdrop of pro-screening discourse [36]. Furthermore, women from low SES and non-White ethnic backgrounds traditionally associated with non-engagement with the NHSBSP [37] were under-represented in our sample and may have different views. Consequently, further research is required to explore the attitudes of ‘hard to reach’ populations towards PRA and RSBS [32].

As women suggest that they will require considerable support throughout the RSBS process, this study mirrors research demonstrating that health professionals are anticipating this screening approach to carry an increased workload [38,39,40]. For instance, high-risk women are anticipated to require emotional and/or practical support in terms of shared decision-making, guidance for risk-management options, and regular reassessment of level of risk. Some women also suggested that if they were assessed as low-risk they would like to have access to breast awareness training, clinical breast examinations and opportunistic screening.

### Strengths and Limitations

As international studies have found cultural variations in women’s responses to the assessment and communication of PRA and RSBS recommendations [19,30], we regard the NHSBSP context of this study as a strength in terms of external validity for informing UK breast screening policy. Further, the one-to-one interview design allowed for a rich and contextualised understanding of variations in women’s thoughts and feelings towards PRA and RSBS. It also provided the opportunity to address participants’ misunderstandings, e.g., conflation of PRA with diagnostic procedures, and thus minimise levels of health worry. There are, however, limitations that should be considered. Although we aimed to provide a comprehensive assessment of attitudes to both PRA and RSBS and explored a range of screening attributes, we used indicative age-ranges and intervals which may not all be realistic for an actual RSBS programme. To avoid cognitive overload, we did not specify quantitative thresholds of risk or explore risk-adapted screening modalities, e.g., the use of ultrasound or MRI for women with high breast density. As these are both important features of RSBS, it is possible their inclusion may have influenced the acceptability of RSBS scenarios. As the study presented RSBS as a hypothetical future approach to screening, it is possible that women will respond differently to a ‘real-life’ change in the NHSBSP. Despite the recruitment of participants with a range of screening experience (including those pre-eligible for screening) and sociodemographic backgrounds, study participants were relatively well-educated, White and not linguistically diverse. Therefore, future studies are required with more diverse samples. It may also be useful to explore the impact of health literacy and numeracy (which we did not assess in this study) on understanding and responses to PRA and RSBS.

## 5. Conclusions

This work provides a unique set of women’s thoughts and feelings about RSBS, and our findings are consistent with other studies in terms of high, but not universal, acceptability for this screening approach. Our findings identify several conditions of acceptability and highlight women’s anticipated needs for information and support. Theory informed behavioural science research will therefore continue to be an essential component of the development of future RSBS policy and communication. The typology of women’s ways of responding to RSBS scenarios may offer a useful starting-point for the development and testing of effective strategies for communicating risk-based options tailored to women’s screening values and preferences.

## Figures and Tables

**Figure 1 cancers-13-05813-f001:**
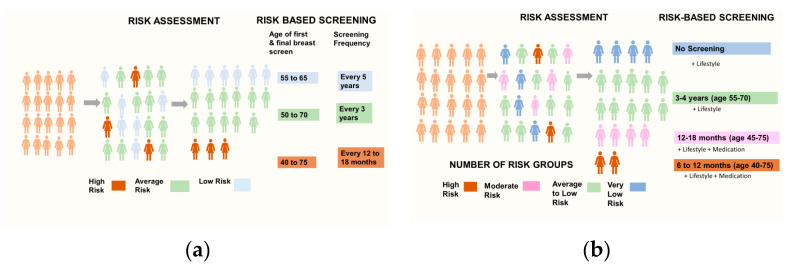
Schematic outline of PRA and RSBS as presented during the interviews. (**a**) Age-range of screening eligibility and screening frequency with three risk groups; (**b**) Age-range of screening eligibility and screening frequency with four risk groups and risk-management options.

**Table 1 cancers-13-05813-t001:** Breast screening experience, sociodemographic and psychosocial characteristics of all participants and those of sub-groups relating to ways of responding: (1)’Overall acceptors’ (2) ‘More is better’ and (3) ‘Screening sceptics’.

Characteristic	All (*n* = 25)	1 Overall Acceptors (*n* = 10)	2 More Is Better (*n* = 9)	3 Screening Sceptics (*n* = 6)
**Age range** (mean, SD)	40–68 (54.96 ± 8.64)	40–68 (53.30 ± 9.62)	41–68 (53.11 ± 8.57)	52–65 (60.5 ± 5.09)
**Age categories**				
40–49	7	3	4	-
50–59	9	4	3	2
60–70	9	2	3	4
**Breast screening experience**				
Pre-eligible	7	4	3	-
Regular attender	7	3	4	-
Occasional attender	6	3	2	1
Non-attender	5	-	-	5
**Occupational social grade**				
AB-C1 (managerial/professional)	13	4	5	4
C2-DE (manual/semi-skilled)	12	6	4	2
**Educational attainment**				
No qualifications	1	-	1	-
GCE/O’level	4	-	4	-
A’level or equivalent	6	2	3	-
University degree	6	4	-	2
Masters or higher	5	1	1	3
Other (e.g., City & Guilds)	3	3	-	1
**Ethnicity**				
White British	19	9	5	5
Black Caribbean	2	-	1	1
Asian	4	1	3	
**Perceived risk of breast cancer**				
Much higher	-	-	-	-
A little higher	4	2	1	1
About the same	14	6	8	1
A little lower	4	1	-	3
Much lower	3	1	-	1
**Breast cancer worry** (intensity)				
Extremely	-	-	-	-
Quite a bit	10	4	6	-
Slightly	10	4	3	3
Not at all	5	2	-	3
**Breast cancer worry** (frequency)				
Very often	-	-	-	-
Often	2	1	1	-
Sometimes	10	6	4	1
Occasional	10	2	4	3
Never	3	1	-	2

**Table 2 cancers-13-05813-t002:** Summary of themes and sub-themes.

Theme	Subtheme
Risk perceptions and acceptability of personalised risk assessment (PRA)	Perceived risk Acceptability of providing risk information Understanding of PRA Willingness to receive risk feedback
Ways of responding to risk- stratified breast screening (RSBS) scenarios	A typology: (1) ‘overall acceptors’; (2) ‘more is better’; (3) ‘screening sceptics’ Age range of screening eligibility Screening frequency No screening—very low risk
Influence of ‘ladder of risk’ on responses to RSBS scenarios	
Concerns and conditions of acceptability	Information support Breast awareness support Integration of PRA and RSBS
Perceived effectiveness: RSBS vs. current NHSBSP	

## Data Availability

The data presented in this study are available on request from the corresponding author. In order to avoid compromising participants’ anonymity, the data are not publicly available.

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
