# Peer review of "UK Women’s Views of the Concepts of Personalised Breast Cancer Risk Assessment and Risk-Stratified Breast Screening: A Qualitative Interview Study"

_cancers, 2021, doi:10.3390/cancers13225813_

Round 1

Reviewer 1 Report

The authors present qualitative findings from interviews with 25 women exploring women's attitudes towards personalised risk assessment (PRA) and risk-stratified breast screening (RSBS). The topic is important and relevant for implementation of personalised medicine in the field of breast cancer.  While I find the topic interesting and relevant, there are a number of issues with the manuscript in its current form. In my opinion, the manuscript requires major revisions and all points must be fully addressed for the manuscript to be considered for publication. I have outlined my concerns in a point-by-point manner below. 

Methods

1. There is no mention of ethics review, informed consent or that the study was conducted in line with the principles of the Declaration of Helsinki. This is a glaring omission and must be clearly addressed.

2. There is no rationale for the use of Framework Analysis. All qualitative methods have relative strengths and limitations. Why is this the most appropriate lens to interpret the qualitative findings?

3. There is no description of the questions used to assess psychological measures pertaining to received breast cancer risk/worry? Were these previously validated questions? Were they developed for the study? Were the questions beta tested? If not previously validated, then this should be noted in the study limitations.

4. The description of the sequence is confusing, i.e., 1) exploring attitudes/awareness/knowledge; 2) cognitive & emotional responses to PRA, 3) exploring benefits 7 harms. This should be edited for clarity to clearly link them to the bulleted aims at the end of the introduction.

5. Figure 1 in the Methods is not clearly described. How were these visualizations developed? Are they existing or were they created for the study? How do we know that the visualizations are understandable to participants. Again this should be noted in the limitations.

Results

6. Table 1 is extremely hard to read and needs to be revised to help the reader. lines run over and parenthetical statements cross lines. Please consider shading or horizontal lines to differentiate sections and please refer to the Journal Guidelines.

7. Please spell out abbreviations at their first use (e.g., IVF, line 156)

8. The reporting of the results leaps directly to the synthesis and does not follow the structure outlined in the Methods. This makes for a confusing presentation. I strongly recommend this be restructured for clarity to avoid simply a laundry list of findings each with a representative quote. 

9. Table 2 is illegible. Why are all the elements center justified? I urge the authors to consider reporting sub themes in a list/bulleted format under each Theme. 

10. The description of how the interview data (collected using the interview guide) was synthesized to identify themes is lacking. How did the authors arrive at these themes? More detail on the process must be provided in the Methods.

11. The Results are lengthy and hard to follow. It would be useful to use tables to facilitate the reader following the findings and major themes. For example, the three identified groups ("Overall acceptors", "More is better", "Screening sceptics") and delineate characteristics and subthemes within each. How many are in each group? this was not reported. 

Discussion

12. The opening paragraph cites another U.K. qualitative study examining PRA & RSBS yet discussion (i.e., compare/contrast) is lacking. This should be added to place the current findings into the context of the existing literature.

13. Lines 492-504 discuss differences in characteristics between the three identified groups yet no statistical analyses are noted in the Methods. This must be clarified and if no formal analysis was performed then the wording should be edited to accurately reflect this.

14. There was not mention of data saturation. This is a critical aspect of qualitative inquiry. How was this assessed - i.e. how do we know that 25 interviews was sufficient to achieve saturation of the concepts being explored in the study?

15. The findings identify a number of important gaps and needs ranging from temporal aspects of ascertaining risk, misconceptions, norms, and self-efficacy. Based on recommendations from the U.K. Medical Research Council on complex interventions, it would seem completely appropriate to note that future work should employ behavioral theory in interventions development. This seems lacking in the manuscript and merits comment and discussion.

16. The description of limitations is insufficient and should be more comprehensive. Beyond the limited sample size, homogeneity, lack of diversity there are many other considerations (as noted above). In particular, a major limitation is that the participants were not assessed for their health literacy/numeracy. This is a real shame as it would offer insights into the three identified groups. 

Author Response

Reviewer: 1

  1. There is no mention of ethics review, informed consent or that the study was conducted in line with the principles of the Declaration of Helsinki. This is a glaring omission and must be clearly addressed.

Response 1: Although Ethical approval had been cited at the end of the  manuscript end under Institutional Review Board Statement. We have now included the following statement within the main text: “The study was approved by the King’s College London research ethics committee  (Ref: LRS-19/20-17949): (Please see p.5, lines: 91-92)

  1. There is no rationale for the use of Framework Analysis. All qualitative methods have relative strengths and limitations. Why is this the most appropriate lens to interpret the qualitative findings?

Response 2: We have included further details as to why Framework Analysis (FA) was considered appropriate (Please see: p.7, lines: 158-199).

  1. There is no description of the questions used to assess psychological measures pertaining to received breast cancer risk/worry? Were these previously validated questions? Were they developed for the study? Were the questions beta tested? If not previously validated, then this should be noted in the study limitations.

Response 3: We used two items from the validated Cancer Worry Scale (CWS: Lerman, Trock, Rimer, et al., 1991). Detail has been added to the manuscript (see p.5, lines: 102-3)

4: The description of the sequence is confusing, i.e., 1) exploring attitudes/awareness/knowledge; 2) cognitive & emotional responses to PRA, 3) exploring benefits-harms. This should be edited for clarity to clearly link them to the bulleted aims at the end of the introduction.

Response 4:  We have made extensive amendments and inclusion of more information detailing the interview procedure in full (pp.5-6, lines: 111-152) to provide extra clarity and coherence to both the Methods and Results sections of the manuscript and ensured consistent reporting between the study aims (as outlined at end of introduction) and study method (see p.5, lines: 111-113 p. 6, lines: 114-121; 122-132 & 146-150). 

5: Figure 1 in the Methods is not clearly described. How were these visualizations developed? Are they existing or were they created for the study? How do we know that the visualizations are understandable to participants.  Again this should be noted in the limitations.

Response 5: Visualisations were designed exclusively for this study (noted on p.6, lines 130-132.). We have also raised the issue of intervention understanding and health literacy within the strengths and limitations section: (p.23, lines: 732-734) Use of a one-to-one interview design enabled provision and discussion of information and clarification of any misunderstandings expressed by the participants. 

  1. Table 1 is extremely hard to read and needs to be revised to help the reader. lines run over and parenthetical statements cross lines. Please consider shading or horizontal lines to differentiate sections and please refer to the Journal Guidelines.

Response 6: We have reformatted and amended Table 1 accordingly.

  1. Please spell out abbreviations at their first use (e.g., IVF, line 156)

Response 7: The text has been edited to include in vitro fertilisation (p. 10, line: 221).

  1. The reporting of the results leaps directly to the synthesis and does not follow the structure outlined in the Methods. This makes for a confusing presentation. I strongly recommend this be restructured for clarity to avoid simply a laundry list of findings each with a representative quote. 

Response 8:   As noted above, considerable additional text has now been included to demonstrate the study interview procedure to add coherence to the method and results sections.

  1. Table 2 is illegible. Why are all the elements center justified? I urge the authors to consider reporting sub themes in a list/bulleted format under each Theme. 

Response 9: We have reformatted and amended Table 2 accordingly to make it easier to read.

  1. The description of how the interview data (collected using the interview guide) was synthesized to identify themes is lacking. How did the authors arrive at these themes? More detail on the process must be provided in the Methods.

Response 10:  As a full description of how we synthesized the data and arrived at the final themes presented in this manuscript would entail adding considerable length, we have included a table outlining  how the preliminary codes and themes were synthesised and appended to the manuscript (please see S3).

  1. The Results are lengthy and hard to follow. It would be useful to use tables to facilitate the reader following the findings and major themes. For example, the three identified groups ("Overall acceptors", "More is better", "Screening sceptics") and delineate characteristics and subthemes within each. How many are in each group? this was not reported. 

Response 11:  We have extensively amended the results to make them easier to follow, adding additional sub-headings, amending Table 2, and including fuller and more contextualised participant quotes.   Table 1 has also been revised to include additional information regarding participant characteristics by response type; ‘overall acceptors’; ‘more is better’ and ‘screening sceptics.

  1. The opening paragraph cites another U.K. qualitative study examining PRA & RSBS yet discussion (i.e., compare/contrast) is lacking. This should be added to place the current findings into the context of the existing literature.

Response 12:  We have edited the text to compare and contrast the two studies (see p.19, lines: 556-561 &  p. 20, lines: 627-29).

  1. Lines 492-504 discuss differences in characteristics between the three identified groups yet no statistical analyses are noted in the Methods. This must be clarified and if no formal analysis was performed then the wording should be edited to accurately reflect this.

Response 13.  As this was a small, qualitative  study and the sample was not designed to be representative, we do not believe that quantitative statistical analyses would be appropriate.  However, the inclusion of participant characteristics by response type in Table 1 provides data to justify our observed difference in characteristics between the three identified groups. We have also amended the wording in the discussion to reflect this (see p.21, lines: 646-661).

 14:  There was not mention of data saturation. This is a critical aspect of qualitative inquiry. How was this assessed - i.e.,  how do we know that 25 interviews was sufficient to achieve saturation of the concepts being explored in the study?

Response 14: We have now noted that we had no a priori sample size and how we recruited women in two batches until it was felt that no new information was forthcoming (please see p.5, lines: 80-92). We do not claim this is ‘data saturation’ however, given the debate in the literature surrounding this concept. (https://www.tandfonline.com/doi/full/10.1080/2159676X.2019.1704846).

  1. The findings identify a number of important gaps and needs ranging from temporal aspects of ascertaining risk, misconceptions, norms, and self-efficacy. Based on recommendations from the U.K. Medical Research Council on complex interventions, it would seem completely appropriate to note that future work should employ behavioural theory in interventions development. This seems lacking in the manuscript and merits comment and discussion.

Response 15:  Thank you very much for this helpful comment. We have amended the concluding section to highlight the important role of health behavioural science (please see p.23, lines:741-743).

  1. The description of limitations is insufficient and should be more comprehensive. Beyond the limited sample size, homogeneity, lack of diversity there are many other considerations (as noted above). In particular, a major limitation is that the participants were not assessed for their health literacy/numeracy. This is a real shame as it would offer insights into the three identified groups. 

Response 16: The limitation section has been expanded and we note that assessment of health literacy/numeracy would be a useful avenue for future research (p.23, lines:732-34).

Reviewer 2 Report

This is a really interesting paper that makes an important contribution to the literature about tailored screening for breast cancer. I make the following suggestions to improve the paper prior to publication:

Introduction needs to take a little more time to explain the specific version of RSBS that is being explored in this paper. the detail is provided in a figure, but it should also be spelt out in the introduction how this will work. for example: will the lifestyle and medication be offered by the screening service? or by referral out? this is different to tailored screening being trialled in other countries, so make it clear what the program will involve.

Methodology requires more detail:

More detail is required about recruitment - who was targeted? how were they invited?

Why was the data collection method of in-depth interviews chosen? who conducted interviews? what was their position/training etc.

Were the different groups of participants analysed together or separately? what was the justification for this decision?

Results are presented with short quotes, which are sometimes hard to fully understand and some need further context. In addition, the typology is interesting, but was not presented fully. I would be interested in more detail about the three groups, what motivated their different approaches? were there any demographic trends? How did they women themselves describe their position? This could be provided before going into detail about how they responded to each facet of risk based screening. A table could help present some of the details about the three groups more concisely. 

Some of the novelty of this study is having the mix of different types of screeners, but more could be made of this. It would be interesting to know more about the non-attenders. How did they, in particular, respond? did they give reasons for prior non-attendance?

Some quotes are concerning - e.g. p8, line 202-204. How did the interviewer respond? were misunderstandings clarified? this ethical issue could be covered in the methods.

Similarly P8 line 167 - this is a really interesting quote - did the interviewer unpack what the participant meant? can more detail be provided? some of the quotes feel quite truncated and de-contextualised, which is not how 'in-depth interview' data is usually presented. In general there seems to be a bit of a mismatch between how the method is described and how the data is presented. 

This could perhaps be resolved by presenting fewer themes, but going into more detail and providing more contextualisation of the data. It would be better if the women's voices, in the context of their lives, came through more strongly.

The discussion appears to introduce new data about the groups that was not presented in the results.  This seems to further confirm the need to cover fewer themes in more detail, but include all relevant results in the results section of the paper. 

Author Response

Reviewer 2

This is a really interesting paper that makes an important contribution to the literature about tailored screening for breast cancer. I make the following suggestions to improve the paper prior to publication:

  • Response: Thank you for recognising the importance of the paper.

Introduction needs to take a little more time to explain the specific version of RSBS that is being explored in this paper. the detail is provided in a figure, but it should also be spelt out in the introduction how this will work. for example: will the lifestyle and medication be offered by the screening service? or by referral out? this is different to tailored screening being trialled in other countries, so make it clear what the program will involve.

  • Response: As no policy has yet been decided on in the UK with regard to RSBS, we have edited the study aims make it clear that we were exploring possible RSBS scenarios rather than a specific version (p.4,lines: 63-65)  We have also detailed how our research focus was on the influence of lifestyle and chemoprevention on the acceptability of risk-stratified screening attributes rather than the acceptability of such risk-management options per see (p.4, lines: 66-67)  which has been studied by other researchers in the RSBS field, e.g., Rainey et al., 2018; 2020).  Furthermore, the theoretical nature of the RSBS model is also now cited as a limitation: (23, lines: 726-728).

Methodology requires more detail: More detail is required about recruitment - who was targeted? how were they invited?

  • Response: We have added further information outlining recruitment procedures (p.5, lines: 79-90).

Why was the data collection method of in-depth interviews chosen? who conducted interviews? what was their position/training etc.

  • Response: have amended the text to explain the rationale for an in-depth interview study design (p.5, lines: 94-97). We have also cited interviewer’s experience and status (p.5, lines: 106-110).

Were the different groups of participants analysed together or separately? what was the justification for this decision?

  • Response: There was no grouping of participants by breast screening or other socio demographic characteristics. The transcripts were analysed together, in order of participation which we have now specified (p.5, lines: 102-104). The different groups of response types emerged from the synthesis stage of data analysis. We then examined the responses to changes in interval, age range the prospect of no screening for each of the groups in the typology. 

Results are presented with short quotes, which are sometimes hard to fully understand, and some need further context. In addition, the typology is interesting, but was not presented fully. I would be interested in more detail about the three groups, what motivated their different approaches? Were there any demographic trends? How did they women themselves describe their position? This could be provided before going into detail about how they responded to each facet of risk based screening. A table could help present some of the details about the three groups more concisely. 

  • Response: Where appropriate, we have now given longer quotes to maintain the context in which they were made (for e.g., please see p.7, lines: 242-243; p.13, lines: 351-354; p.14, lines: 377-380; 389-394). For further detail about participant characteristics relating to each identified response type, please see amendments to Table 1 which now includes three additional columns describing participant characteristics by response type: ‘overall acceptors’; ‘more is better’ and ‘screening sceptics’.

Some of the novelty of this study is having the mix of different types of screeners, but more could be made of this. It would be interesting to know more about the non-attenders. How did they, in particular, respond? Did they give reasons for prior non-attendance?

  • Response: We have woven in more detail regarding motivations to attend screening or not within the results section. For example, please see pp.10-11, (lines: 237-244), where we cite non-attenders concerns about radiation and screening-related harms.

Some quotes are concerning – e.g., p. 8, line 202-204. How did the interviewer respond? Were misunderstandings clarified? This ethical issue could be covered in the methods.

  • Response: From an ethical perspective, we don’t feel that anyone was unduly alarmed by the interview as (a) most women perceived the approach to be an improvement and (b) no women accepted the offer of links to emotional support at close of the interview. Some women did misinterpret information e.g., equating a high PRA result with a cancer diagnosis. Accurate information was reinforced throughout the interview after exploring women’s views. These misunderstandings are important as they illustrate potential responses to risk information in a real-life scenario. The fact that the interview was purely hypothetical and women were not being given personalised risk information minimised any risk of harm. We have also referred to the use of a one-to-one interview design as an opportunity to address participants’ misunderstanding in Strengths and Limitations (please see p.23,  lines: 716-720). 

Similarly P8 line 167 - this is a really interesting quote - did the interviewer unpack what the participant meant? can more detail be provided? some of the quotes feel quite truncated and de-contextualised, which is not how 'in-depth interview' data is usually presented. In general, there seems to be a bit of a mismatch between how the method is described and how the data is presented. This could perhaps be resolved by presenting fewer themes, but going into more detail and providing more contextualisation of the data. It would be better if the women's voices, in the context of their lives, came through more strongly.

  • Response: We don’t feel it is appropriate to remove themes but as noted above we have provided fuller quotes to provide more context.

The discussion appears to introduce new data about the groups that was not presented in the results.  This seems to further confirm the need to cover fewer themes in more detail, but include all relevant results in the results section of the paper. 

  • Response: Our amendments to Table 1 will help to clarify the differential characteristics of each response type to RSBS proposals and thus present the data highlighted in the discussion without the need to remove themes, all of which we feel important to include to reflect the range of views and responses raised in the interviews.

Reviewer 3 Report

  1. It should be included in the title that the article refers to UK women only. Please correct.
  2. Figure 1, please modify the colors in the Figure, since the lightest color used is not visible in the printed text.

Author Response

Reviewer 3

  1. It should be included in the title that the article refers to UK women only. Please correct.

  • Response: We have amended the study title to specify that this study concerns UK participants.

2: Figure 1, please modify the colors in the Figure, since the lightest color used is not visible in the printed text.

  • Response: We have adjusted the colour tone for Figure 1.

Round 2

Reviewer 1 Report

The authors have responded to the points raised in the initial review. I found the revised manuscript to be more coherent and stronger. I feel the manuscript is suitable for publication but recommend spelling our "UK" in the title.

Reviewer 2 Report

the authors have addressed the concerns I raised and I think the paper is now suitable for publication